# It Looks Like a Spinal Cord Tumor but It Is Not

**DOI:** 10.3390/cancers16051004

**Published:** 2024-02-29

**Authors:** Julien Fournel, Marc Hermier, Anna Martin, Delphine Gamondès, Emanuele Tommasino, Théo Broussolle, Alexis Morgado, Wassim Baassiri, Francois Cotton, Yves Berthezène, Alexandre Bani-Sadr

**Affiliations:** 1Department of Neuroradiology, East Group Hospital, Hospices Civils de Lyon, 59 Bd Pinel, 69500 Bron, France; julien.fournel@chu-lyon.fr (J.F.); marc.hermier@chu-lyon.fr (M.H.); anna.martin@chu-lyon.fr (A.M.); delphine.gamondes@chu-lyon.fr (D.G.); emanuele.tommasino@chu-lyon.fr (E.T.); yves.berthezene@chu-lyon.fr (Y.B.); 2Department of Spine and Spinal Cord Neurosurgery, East Group Hospital, Hospices Civils de Lyon, 59 Bd Pinel, 69500 Bron, France; theo.broussolle@chu-lyon.fr (T.B.); alexis.morgado@chu-lyon.fr (A.M.); wassim.baassiri@chu-lyon.fr (W.B.); 3CREATIS Laboratory, CNRS UMR 5220, INSERM U1294, Claude Bernard Lyon I University, 7 Avenue Jean Capelle, 69100 Villeurbanne, France; francois.cotton@chu-lyon.fr; 4Department of Radiology, South Lyon Hospital, Hospices Civils de Lyon, 165 Chemin du Grand Revoyet, 69495 Pierre-Bénite, France

**Keywords:** spinal cord, magnetic resonance imaging, myelopathy, neoplasms, inflammatory pseudotumor

## Abstract

**Simple Summary:**

This review delineates the diagnostic challenges in distinguishing neoplastic from non-neoplastic spinal cord pathologies, highlighting the importance of a comprehensive radiological evaluation. An integral component of this evaluation is the detailed analysis of MRI findings to accurately diagnose lesions that mimic spinal cord tumors. It emphasizes the need for careful consideration of common non-surgical myelopathies in differential diagnoses due to their higher prevalence. Additionally, the review discusses the principal etiologies of spinal pseudotumors, including inflammatory, vascular, and infectious neurological diseases. This approach aims to refine diagnostic accuracy and enhance clinical decision-making by providing a nuanced understanding of the varied manifestations of spinal cord pathologies.

**Abstract:**

Differentiating neoplastic from non-neoplastic spinal cord pathologies may be challenging due to overlapping clinical and radiological features. Spinal cord tumors, which comprise only 2–4% of central nervous system tumors, are rarer than non-tumoral myelopathies of inflammatory, vascular, or infectious origins. The risk of neurological deterioration and the high rate of false negatives or misdiagnoses associated with spinal cord biopsies require a cautious approach. Facing a spinal cord lesion, prioritizing more common non-surgical myelopathies in differential diagnoses is essential. A comprehensive radiological diagnostic approach is mandatory to identify spinal cord tumor mimics. The diagnostic process involves a multi-step approach: detecting lesions primarily using MRI techniques, precise localization of lesions, assessing lesion signal intensity characteristics, and searching for potentially associated anomalies at spinal cord and cerebral MRI. This review aims to delineate the radiological diagnostic approach for spinal cord lesions that may mimic tumors and briefly highlight the primary pathologies behind these lesions.

## 1. Introduction

Differentiating neoplastic from non-neoplastic spinal cord pathologies is challenging due to similar clinical and radiological features. Spinal cord tumors, representing just 2–4% of central nervous system tumors, are less common than non-tumoral myelopathies, which typically have inflammatory, vascular, or infectious origins [1]. Spinal cord biopsies carry a risk of worsening neurological function and may produce up to 30% false negatives or misdiagnoses [2,3]. Therefore, more common causes of myelopathy-mimicking tumors, which usually do not require surgery, should be considered to avoid unnecessary and potentially harmful interventions. This article aims to delineate the radiological diagnostic approach for spinal cord lesions that may mimic tumors and to highlight the primary pathologies behind spinal cord pseudotumors.

## 2. Radiological Diagnostic Approach

### 2.1. Detecting the Spinal Cord Lesion

First, the detection of the spinal cord lesion mainly relies on MRI, primarily with T1-weighted imaging (T1WI) and T2-weighted imaging (T2WI), complemented by post-contrast T1WI. Additional MRI sequences like gradient-recalled echo T2*-weighted imaging (T2*-WI) or susceptibility-weighted imaging, diffusion-weighted imaging (DWI), or T1WI or T2WI with fat signal saturation (FAT-SAT) can provide additional insights. In cases of suspected degenerative compressive myelopathy (DCM), dynamic MRI might uncover compression not visible at static MRI [4]. Clinical use of diffusion tensor-imaging tractography is limited by technical challenges, owing to the small size of spinal cord, and cerebrospinal fluid (CSF) artifacts [5]. Despite its poor contrast resolution, CT myelography serves as an alternative when MRI is contraindicated and may identify spinal cord enlargement, thereby locating the lesion [6]. CSF analysis also contributes, especially in detecting inflammatory changes or abnormal cells such as lymphomatous or metastatic cells.

### 2.2. Localizing the Spinal Cord Lesion

The second stage focuses on precisely localizing lesions and distinguishing intra-medullary masses from intradural extramedullary and extradural masses. Sagittal MRI of intra-medullary lesions typically reveals spinal cord enlargement without notable displacement. The extent of the lesion is a critical feature to consider. Lesions spanning three or more vertebral segments are longitudinally extensive; others are considered as short. Longitudinally extensive myelitis is a classic finding in various pathologies such as neuromyelitis optica syndrome disorders (NMOSD), either mediated by myelin-oligodendrocyte glycoprotein antibody (MOGAD) or aquaporine-4 antibody (AQP4+), neurosarcoidosis, anti-glial fibrillary protein (GFAP) encephalomyelitis, infectious myelitis, or connective tissue diseases [7,8,9,10]. Conversely, short lesions are frequent findings in multiple sclerosis (MS) [11]. Axial MRI assessment provides additional insights. Lesions affecting both halves of the spinal cord are classified as transverse. Transverse myelitis may arise from various etiologies including acute infection, acute disseminated encephalomyelitis (ADEM), demyelinating inflammatory myelitis, or in systemic lupus erythematosus [12]. The selective involvement of the grey and white matter may offer additional insights. Generally, MS affects white matter and spares grey matter, while selective gray matter involvement, resembling an H-shape, is seen in MOGAD [7,11]. Yet, conditions like spinal cord infarction or viral myelitis may present similarly [13,14,15]. Anterior horn “polio-like” involvement may suggest pathologies such as anterior spinal artery infarct, chronic compressive myelopathy, or viral myelitis [16,17,18]. The level of spinal cord involvement is a less reliable indicator since both tumors and pseudotumors may involve either cervical or thoracic segments. However, inflammatory myelopathies more commonly affect cervical segments, whereas conus involvement can indicate conditions like MOGAD or spinal dural arteriovenous fistula (SDAVF) [7,19].

### 2.3. Assessing the Lesion’s Signal Intensity Characteristics

The third step assesses lesion signal intensity, especially T1 and T2 signals, and features like enhancing areas, cystic components, or diffusion-restricted zones. This involves the assessment of T1 and T2 signal intensity, and identifying enhancing portions, cystic components, or areas of diffusion restriction. Emphasis should be on T2WI hypointense lesions indicative of hematomyelia, potentially caused by tumoral bleeding, idiopathic factors, spinal vascular malformations, coagulation disorders, or radiation-induced changes. While this could result from tumoral bleeding, other etiologies include idiopathic causes, spinal vascular malformations, coagulations disorders or related to radiation-induced changes [20,21]. If both T1WI and T2WI are hyperintense, with T2WI FAT-SAT showing hypointensity, it often indicates fatty tissue, a hallmark of lipomas [22]. Bright spotty lesions, defined by T2 signal intensity equal to or surpassing that of CSF, are a hallmark of AQP4 + NMOSD, though it may also be encountered in spinal cord infarction [23,24]. Cystic lesions, which can range from fluid-filled to protein-rich, often suggest pathologies such as syrinx, dermoid, or epidermoid cysts, among other developmental anomalies. Diffusion restriction on MRI is a key feature, often seen in pyogenic abscesses, spinal cord infarctions, or hypercellular neoplasms like lymphoma [10,25,26]. The definition of lesion boundaries also offers diagnostic insights. Cavernomas generally have well-defined, regular margins, whereas spinal astrocytoma and inflammatory myelitis, such as NMOSD and neurosarcoidosis, may exhibit poorly defined boundaries [27]. A specific feature to note is the “cap sign”, a distinctive hypointense rim on T2*WI, often seen in pathologies like ependymomas or cavernomas [28,29].

### 2.4. Evaluating the Lesion Environment and Associated Anomalies in Spinal Cord MRI

The fourth step involves analyzing associated anomalies in spinal cord MRI. Multiple, asymmetrical lesions with increased T2 signal often suggest chronic inflammatory demyelinating diseases like MS [11]. Features like spinal central canal enhancement might indicate neurosarcoidosis [9] or MOG. Pseudodilatation of the spinal central canal, observed frequently in MOGAD, is notably absent in MS myelitis [30]. Tortuous and dilated perimedullary vessels indicate a spinal dural arteriovenous fistula (SDAVF) [19]. Assessing the adjacent bony anatomy can also be revealing since vertebral scalloping might indicate slow-growing lesions such as myxopapillary ependymomas [31]. Canal stenosis or alignment of the spinal cord with disc protrusions are supportive of a diagnosis [32,33]. Identifying defective fusion in vertebral posterior elements, a sign of spinal dysraphism, can support congenital malformation diagnoses such as lipomas, dermoid, or epidermoid cysts [34].

### 2.5. Evaluating Associated Anomalies at Cerebral MRI

The fifth step involves assessing potentially associated cerebral abnormalities via cerebral MRI that may be useful in excluding differential diagnoses such as encephalomyelitis. Multiple, small, round hyperintense lesions on T2/fluid-attenuated inversion recovery (FLAIR) sequences in brain MRI, particularly located in periventricular, subcortical, or posterior fossa regions, strongly indicate MS [11]. Bilateral optic neuritis involving the chiasma, focal lesions in areas like the area postrema, midbrain, or diencephalon and “pencil-thin” subependymal enhancement are characteristic of AQP4 + NMOSD [35,36]. Unilateral or bilateral optic neuritis with perineural enhancement and extensive cerebellar lesions, especially across the cerebellar peduncles, may be indicative of MOGAD [7]. Pachy- or leptomeningeal enhancement of the basal cranium indicates conditions like neurosarcoidosis, neuro-Behçet’s disease, and neurotuberculosis [37,38]. ADEM typically manifests with asymmetrical, small, punctate-to-tumefactive lesions in both supratentorial and cerebellar regions [39]. Herpes virus encephalitis typically shows edematous or necrotic lesions with diffusion restriction in the internal temporal lobes, cingulum, and insula [40]. Anti-GFAP encephalomyelitis is characterized by T2/FLAIR hyperintensities in the semi-oval center white matter, accompanied by radial perivascular contrast enhancement [41].

### 2.6. Features Distinguishing Spinal Cord Tumors from Pseudotumors

Comparative studies exploring the clinical and radiological characteristics of actual spinal cord tumors versus mimicking pathologies are limited. A key study investigating the histopathological outcomes in 212 patients undergoing surgery for suspected spinal cord tumors found a 4% incidence of pseudotumors [42]. The primary pathologies behind these pseudotumors were predominantly inflammatory myelitis, neurosarcoidosis, and amyloid angiopathy, with spinal cord enlargement less commonly observed in these cases compared to actual tumors [42]. A more recent analysis of 43 patients initially suspected of having spinal cord tumors revealed that astrocytoma was the most frequent initial diagnosis [43]. However, later findings revealed that 80% of these cases were inflammatory myelitis and 20% were DCM (with shorter symptom duration and less frequent spinal cord enlargement than typical astrocytoma cases) [43]. The study further emphasized that most of the pseudotumors manifested as short lesions [43]. In clinical practice, primary spinal cord tumors like astrocytomas and ependymomas are often associated with total disappearance of CSF spaces nearby at diagnosis due to frank cord enlargement, whereas some CSF may still be visible around the spinal cord in several pseudotumoral lesions. However, this finding may be lacking in small tumoral lesions including hemangioblastomas, intramedullary metastases, or lymphomas. From a clinical perspective, this study indicated that patients with neurological diseases had significantly shorter symptom duration than those with neoplasms [43]. In patients presenting with acute spinal cord swelling, corticosteroid therapy could help differentiate pseudotumors from true tumors, although infection must be ruled out beforehand [42]. Additionally, early MRI monitoring at 6 weeks could be beneficial in these patients or in cases of atypical clinical presentation.

These findings highlight the importance of understanding primary pathologies that mimic spinal cord tumors, especially in acute stages.

## 3. Main Pathologies Mimicking Spinal Cord Tumors

### 3.1. Inflammatory Myelitis

#### 3.1.1. Short Myelitis

##### Multiple Sclerosis

MS is a neuroinflammatory demyelinating disease of the central nervous system and one of the leading causes of disability among young adults [11]. Aside from idiopathic myelitis, MS is the predominant cause of inflammatory myelopathy [44]. About 90% of MS patients show spinal cord lesions with typical findings being multiple T2WI hyperintense lesions [11,16]. In MS, typical findings are multiple T2WI hyperintense lesions primarily affecting the cervical spine, dorsal and lateral columns [16]. Acute MS myelitis (Figure 1) tends to be short, spanning over less than three vertebral segments and occupying less than half of a transverse section, with associated edema and mild spinal cord enlargement [16]. Enhancement patterns in MS vary from non-specific to nodular or patchy, with annular enhancement being strongly suggestive [16]. It usually resolves within two months, though T2 signal abnormalities may persist [16]. Usually, short and multiple spinal cord lesions in MS do not pose a differential diagnosis problem with tumors. In some instances, MS may occasionally cause longitudinally extensive myelitis, especially in children [27]. The spatial dissemination and coexistence of brain lesions of different ages are major clues to diagnosis [11]. Brain lesions are typically small, round, or oval and may selectively involve the white matter in periventricular, subcortical, and posterior fossa regions [11].

##### Other Inflammatory Causes of Short Myelitis

Progressive solitary sclerosis and pure relapsing short myelitis, conditions related to MS, feature a single spinal lesion without cerebral involvement thereby not meeting the required spatial dissemination criteria for MS. It requires at least two episodes of myelitis, typically short, and negative antibody tests [27]. The lesions more commonly affect the cervical spine and peripherally posterior regions [27]. Lesions are usually enhanced [27].

A diagnosis of short inflammatory myelitis does not exclude other conditions. Short lesions might be related to other demyelinating inflammatory diseases, which usually cause longitudinally extensive myelitis.

#### 3.1.2. Longitudinally Extensive Myelitis

Longitudinally extensive inflammatory myelitis comprises a wide range of pathologies including AQP4 + NMOSD, MOGAD, neurosarcoidosis, ADEM, or neuro-Behçet’s disease. Less commonly, autoimmune myelitis and connective tissue disorders such as systemic lupus erythematosus may be associated.

##### Neuromyelitis Optica Spectrum Disorders Positive for Aquaporin-4 IgG

AQP4 + NMOSD is a primary cause of longitudinally extensive myelitis. Up to 90% of spinal cord involvement in AQP4 + NMOSD consists in longitudinally extensive transverse myelitis [45]. Characteristic acute lesions (Figure 2) typically show spinal cord enlargement and concurrent T1WI hypointensity, and an almost systematic contrast enhancement [46]. Longitudinal leptomeningeal enhancement, typically at the level of parenchymal enhancement, is a common associated feature [47]. Bright spotty lesions are a classical finding in AQP4 + NMOSD [23]. Incomplete ring enhancement, a specific imaging pattern, may also be observed [48]. In the brain, AQP4 + NMOSD typically presents with bilateral optic neuritis affecting the chiasma and focal lesions in specific areas, including the area postrema, midbrain, and diencephalon [49]. Additionally, a “pencil-thin” ependymal enhancement is highly indicative of the disorder [35].

##### Myelin-Oligodendrocyte Glycoprotein Antibody Associated Disease

In MOGAD, myelitis is observed in approximately 25% of adult patients and presents as longitudinally extensive in 60–80% of these cases [50]. The lesions are more commonly found in the conus medullaris [51]. During the acute phase, enhancement of lesions is infrequent, while leptomeningeal enhancement is more commonly observed, potentially accompanied by diffuse enhancement and thickening of nerve roots [52,53,54,55]. Centrally located lesions often show a linear appearance on sagittal MRI, described as the “ventral sagittal line” [7]. Axial imaging reveals that, in addition to white matter involvement, gray matter lesions may form an H-shaped pattern, indicative yet not exclusively specific to MOGAD [7]. MOGAD myelitis (Figure 3) often shows “pseudodilatation” of the spinal central canal, a feature absent in MS myelitis [54], and also enhancement in the dilatation of the spinal central canal. Characteristic brain MRI findings in MOGAD include cerebellar lesions, extending to the cerebellar peduncles, and optic neuritis with perineural enhancement, which can be either unilateral or bilateral [56].

##### Neurosarcoidosis

In neurosarcoidosis, about one-quarter of patients develop longitudinally extensive myelitis, primarily affecting the cervical and high thoracic spinal cord [37]. In the acute phase (Figure 4), enhancement, usually linear in the dorsal subpial white matter or less commonly ventral, is almost always present [9]. It is typically linear in the dorsal subpial white matter, or less commonly ventral [9]. Ependymal enhancement, when associated, may result in a characteristic “trident sign” on axial images [57]. Multiple enhancing spinal cord micronodules may sometimes occur. Dural masses and leptomeningeal enhancement of the cauda equina nerve roots are sometimes encountered [9]. Although variable, associated cerebral lesions such as meningeal enhancement, often at the base of the skull and cranial nerves, or intra- or extra-axial masses, may be present [37].

##### Acute Demyelinating Encephalomyelitis

ADEM, typically following viral infection or vaccination, predominantly affects children and adolescents [58]. Longitudinally extensive myelitis occurs in a third of patients [58]. Radiologically, it may result in transverse myelitis sometimes accompanied by enhancement (Figure 5) [59]. On brain MRI, multiple small punctate lesions or moderately swelling extensive lesions in T2/FLAIR hyperintensity of the white matter are commonly observed at both supratentorial and infratentorial levels [58]. The lesions are generally bilateral but asymmetric, with possible but inconsistent involvement of the cerebral cortex and basal ganglia [58].

##### Neuro-Behçet’s Disease

Neuro-Behçet’s disease encompasses the neurological manifestations of Behçet’s disease, a chronic inflammatory vasculitis of unknown etiology, predominantly affecting males from the Mediterranean basin, the Middle East, and East Asia [60]. Though uncommon, myelitis is more commonly longitudinally extensive (Figure 6) [61]. Axial MRI typically shows spinal cord lesions with peripheral T2WI hyperintensity and central T2WI hypointensity, creating a “bagel-like” appearance [62]. Cerebral involvement may include pseudotumoral lesions in the brainstem, basal ganglia, and subcortical matter, with leptomeningeal enhancement in one-third of cases [38]. Cerebral vein thrombosis is frequent.

##### Anti-GFAP Encephalomyelitis

Discovered in 2016, anti-GFAP encephalomyelitis is an astrocytopathy that may result in an acute meningoencephalomyelitis [41]. It predominantly affects individuals over 40 years of age, with a slight female predominance [63]. Myelopathy occurs in 27 to 68% of patients [64,65]. MRI typically reveals hazy, ill-defined, longitudinally extensive T2WI lesions in the spinal cord with punctate, central canal, and leptomeningeal enhancement [66]. Cerebral involvement is almost constant, characterized by T2/FLAIR hyperintensities in the white matter of the semioval centers, accompanied by suggestive radial perivascular contrast enhancement [41].

##### Autoimmune Myelitis

Autoimmune myelopathies, rare and often part of multifocal paraneoplastic syndromes, are caused by autoantibodies specific to the central nervous system [27]. With the advent of immunotherapy in the treatment of systemic cancers, their incidence is expected to increase [67]. The most commonly associated cancers are breast and lung carcinomas [67]. Antibodies against the glycine receptor are typically associated with autoimmune myelitis [68]. Other antibodies involved include amphiphysin and collapsin response mediator protein 5-IgG [69]. Spinal cord lesions are generally longitudinally extensive and consist of symmetric and specific-tracts T2WI hyperintensities [67]. Patchy enhancement is observed in about half of the cases [67]. Forms of acute necrotizing longitudinally extensive myelitis have also been described [70].

### 3.2. Degenerative Compressive Myelopathy

DCM is recognized as the primary cause of non-traumatic spinal cord dysfunction globally [71]. DCM typically exhibits chronic myelomalacic changes, characterized by increased T2 signal intensity and decreased T1 signal intensity in segments of the spinal cord under compression [72,73]. Notably, T2 signal abnormalities may selectively affect the ventral horns [18]. The absence of contrast enhancement in DCM may be useful to distinguish it from neoplastic or demyelinating diseases [74]. However, up to 7% of cases may demonstrate enhanced lesions which are thought to result from a breakdown of the blood–spine barrier [75]. DCM can also present acutely as myelitis [76]. DCM lesions range from longitudinally extensive to short, usually showing fusiform T2WI hyperintensity with blurred borders and spinal cord enlargement. Post-contrast sagittal imaging often reveals the “pancake-like sign”, a transverse band of enhancement equal in width and height, located at or just below the maximal compression point of the spinal cord (Figure 7) [77]. On axial views, this manifests as a ring of contrast enhancement encircling the peripheral white matter, sparing the central gray matter [77]. Dynamic MRI, particularly in flexion and extension, can be useful in detecting occult compression, allowing for the assessment of dynamic spinal canal changes including cervical spine motion, alterations in disc positions, variations in spinal canal diameter, and foraminal narrowing [78].

### 3.3. Vascular Myelopathies

#### 3.3.1. Spinal Cord Infarction

Spinal cord infarction, a cause of acute myelopathy, often presents diagnostic challenges in the absence of suggestive contexts, such as aortic surgery [13]. It is reported to affect 14–16% of patients evaluated for transverse myelitis [13,79]. Anterior spinal artery infarctions are the most common [80]. The spinal cord lesions may be short or long, predominantly affecting the gray matter. On T2WI, a “pencil-like” appearance on sagittal views and an “owl-eye” appearance on axial views are indicative but not specific [13]. Other T2WI abnormalities may include punctate gray matter lesions, H-shaped lesions involving the entire gray matter, or transverse holocord involvement (Figure 8) [13]. Diffusion-weighted imaging (DWI) demonstrating diffusion restriction is crucial for confirming the diagnosis of acute spinal cord infarction (Figure 9) [13]. Additional supportive signs of spinal cord infarction include an association with adjacent vertebral body infarction (uncommon), contrast enhancement showing cranio-caudal band-like patterns on sagittal images, and predominance in the gray matter on axial images. Cervical spinal cord infarction may be due to a vertebral artery arterial dissection [13].

#### 3.3.2. Spinal Dural Arteriovenous Fistula

SDAVFs are the most prevalent spinal vascular malformations, constituting 70% of all such entities [81]. They predominantly affect men over 50 years of age and manifest as progressive lower back pain and lower limb weakness [81]. Associated myelopathies (Figure 10), usually longitudinally extensive and involving the lower thoracic region and conus medullaris, typically show T1WI hypointensity, T2WI hyperintensity, and often irregular enhancement [19]. The level of the arteriovenous shunt does not correlate with the level of myelopathy. Tortuous, dilated vessels appearing as signal voids on T2WI and enhanced on post-contrast T1WI strongly indicate this condition. While CT angiography detects the fistulous point in 75% of cases, digital subtraction angiography remains the diagnostic gold standard [19]. This procedure necessitates catheterization of the entire intercostal, lumbar, median and lateral sacral, vertebral, ascending cervical, and even intracranial arteries if no fistula is detected [19]. It may be lengthy, radiating, and catheterization can be made difficult by aortic atheroma in elderly subjects. Consequently, intra-aortic CT angiography has been proposed to study the angioarchitecture of arteriovenous malformations and to locate the artery of Adamkiewicz [82].

While rare, intracranial dural arteriovenous fistulas with perimedullary venous drainage (Cognard type V arteriovenous fistulas) may also manifest as rapid and progressive myelopathy [83].

#### 3.3.3. Spinal Cavernous Malformation

Accounting for about 5% of adult intramedullary lesions, spinal cavernous malformations predominantly appear at cervico-thoracic levels [84]. Usually, without recent major hemorrhage, they manifest as short lesions with slight spinal enlargement. On MRI (Figure 11), these malformations appear as oval-shaped lesions with well-defined and regular margins [85]. They exhibit heterogeneous T1WI and T2WI signals due to the presence of blood products of varying ages and may show complete T2*WI hypointensity [85]. The appearance of a T2WI or T2*WI hypointense rim is suggestive [29]. The presence of associated enhancement is rare and, if present, should primarily raise suspicion for a neoplastic lesion, such as a hemorrhagic ependymoma or a melanoma metastasis. 

#### 3.3.4. Hematomyelia

Hematomyelia may require investigating secondary factors in non-traumatic cases. The leading cause of hematomyelia is spinal trauma, followed by vascular and cavernous malformations in non-traumatic etiologies [86]. The signal intensity of hematomyelia (Figure 12) varies with its chronologic progression [87]. If no underlying lesion is identified initially, a follow-up MRI after 3 months is recommended to ensure that no lesion is obscured by the hematoma. Hemorrhagic cavernomas usually shrink at follow-up, whereas tumors do not.

### 3.4. Infectious Myelopathies

#### 3.4.1. Bacterial Myelitis

Rare in both adults and children, intraspinal abscesses are typically linked to bacterial meningitis, spondylodiscitis, or infective endocarditis [88]. In children, spinal abscesses warrant checking for underlying dysraphism or dermal sinus [89]. *Staphylococcus* and *Streptococcus* predominantly cause intramedullary abscesses [90]. Clinically, they present as a febrile spinal deficit with associated back pain, developing subacutely. The spinal abscess manifests as an expansive intramedullary lesion, with peripheral annular enhancement. On imaging, it appears as T1WI hypointense and T2WI hyperintense, accompanied by perilesional spinal cord edema. Diffusion sequences in pyogenic abscesses show hyperintensity with reduced ADC, like they appear in the brain [90]. Diffusion-weighted imaging is crucial in differentiating necrotic tumor lesions from pyogenic spinal abscesses, with the latter showing hyperintensity on diffusion and significant restriction on the ADC map [91].

Uncommon bacterial pathogens (Figure 13) include syphilis, neuroborreliosis, and tuberculosis. Historically, syphilis was a leading cause of myelopathy before the advent of antibiotics [88]. The classic form, tabes dorsalis, involves degeneration of the posterior columns and dorsal roots at the lumbosacral and lower thoracic levels [92]. Acute myelitis is also possible, displaying a characteristic “candle guttering” appearance and a “flip-flop” sign [93]. Myelitis due to neuroborreliosis lacks specific MRI characteristics but tends to occur more frequently at the cervical level [88]. Tuberculosis may occasionally present as myelitis or intraspinal tuberculomas [88]. Tuberculous myelitis typically affects the thoracic level and is characterized by T2WI hyperintensity, iso- or hypointensity on T1WI, and variable enhancement [94]. Tuberculomas may appear as hypointense T2 focal lesions.

#### 3.4.2. Viral Myelitis

Viral myelitides are most commonly caused by herpesviruses, polio-like viruses, flaviviruses, human immunodeficiency virus, and human T-lymphotropic virus-1 [88]. MRI manifestations of these infections are diverse, ranging from short-segment transverse myelitis to longitudinally extensive forms [88]. Polio-like viruses, including enteroviruses, may preferentially affect the anterior horns, leading to an “owl-eye” pattern, while flaviviruses like West Nile virus may selectively affect the gray matter, resulting in an “H-shape” on axial MRI images [15,27]. Enhancement patterns are inconsistent, varying from diffuse to patchy or peripheral, and may be accompanied by leptomeningeal enhancement and radiculitis [88]. Involvement of the dorsal root ganglia is indicative of herpesviruses infection [95]. Definitive diagnosis largely relies on polymerase chain reaction testing of CSF.

#### 3.4.3. Parasitic Myelitis

Parasitic myelitis, though rare outside endemic regions, can arise from infections such as cysticercosis, schistosomiasis, and echinococcosis. Schistosomiasis (Figure 14) is a trematode worm infection prevalent in endemic areas of Africa, South America, and Asia. Neurological involvement in schistosomiasis is uncommon and primarily affects the spinal cord. Spinal MRI typically reveals a swelling, poorly defined intramedullary lesion, often situated at the conus medullaris [96]. Imaging features include T1WI isointensity, T2WI hyperintensity, and patchy or nodular enhancement post-gadolinium injection, usually indicating a non-extensive myelitis [97]. A travel history to or residing in an endemic region, combined with myelitis of the conus medullaris, should raise suspicion for schistosomiasis.

#### 3.4.4. Fungal Myelitis

Fungal myelitis is a rare condition that primarily affects immunocompromised patients. Etiologies include coccidioidomycosis, histoplasmosis, and cryptococcosis [88]. These infections commonly manifest as acute necrotizing conditions [88].

### 3.5. Metabolic and Toxic Myelopathies

#### Subacute Combined Degeneration of the Spinal Cord and Mimics

Vitamin B12 (cobalamin) deficiency may cause subacute combined degeneration of the spinal cord [98]. Typical clinical manifestations include bilateral and symmetrical sensory disturbances such as hypoesthesia, dysesthesia, and gait abnormalities. Cervical and upper thoracic are the primary affected segments [98]. Subacute combined degeneration of the spinal cord and mimics (Figure 15) present as longitudinally extensive T2WI hyperintense lesions symmetrically affecting the dorsal columns, or less commonly the lateral columns, resulting in a “chevron” or “inverted V pattern” on axial images [98]. Generally, spinal enlargement is mild, and lesions typically do not enhance, although cases with moderate enhancement have been reported [98].

Nitrous oxide toxicity manifests with radiological findings that are identical to cobalamin deficiency and predominantly affects dentists, as well as medical and nursing staff [98]. This condition should be suspected when symptoms arise post-surgically, with the understanding that these symptoms may be delayed in their onset following exposure [98].

Copper deficiency clinically and radiologically mimics B12-deficiency-induced myelopathy, making them indistinguishable [99]. Similarly, methotrexate-induced myelopathy can mimic the radiological appearance of B12 and copper deficiencies [100]. This less common condition than methotrexate-induced leukoencephalopathy arises post-intrathecal administration [100].

Other causes of metabolic and toxic myelopathies include heroin toxicity and excessive intake of vitamin B6 (pyridoxine) [98].

### 3.6. Congenital Spine and Spinal Cord Malformations

#### 3.6.1. Hydromyelia and Syringomyelia

Enlargement of the central canal is defined as hydromyelia, while cystic formations located externally to the central canal are identified as syringomyelia [101]. These conditions often present similarly as fluid-filled central lesions within the spinal cord [101]. Syringomyelia is frequently associated with congenital anomalies such as Chiari I malformation or tethered cord syndrome and may also arise secondary to trauma, meningitis, or neoplasms [101]. Initial diagnostic approaches include comprehensive spinal imaging with contrast to assess for the presence of neoplasms, posterior fossa anomalies, or evidence of tethered cord syndrome [101]. Lesions situated at or proximal to the conus medullaris are referred to as the terminal ventricle [102].

#### 3.6.2. Spinal Dysraphisms

Closed spinal dysraphisms encompass a diverse spectrum of conditions that may manifest through expansive intraspinal lesions [102]. Within this category, intramedullary lipomas are occasionally observed. Lipomas (Figure 16) are distinguished by a well-defined solid mass composed of adipose tissue. Beyond lipomas, the spectrum includes epidermoid and dermoid cysts [102]. Intramedullary epidermoid cysts, although exceedingly rare, characteristically exhibit high signal intensity on T2-weighted imaging, low signal intensity on T1-weighted imaging, and lack of contrast enhancement [103]. A distinctive feature of these cysts is the presence of diffusion restriction on DWI. In contrast, dermoid cysts do not demonstrate diffusion restriction and contain heterogeneous content, including fatty components, typically without contrast enhancement. Rupture of these cysts (Figure 17) may lead to aseptic meningitis [103].

Figure 18 presents the primary causes of spinal cord pseudotumors, while Table 1 details the key radiological characteristics of the most frequent myelopathies discussed.

## 4. Discussion

It can be difficult for clinicians to differentiate between neoplastic and non-neoplastic lesions within the spinal cord. The diagnostic management requires a meticulous evaluation of imaging data. After detection, precise lesion localization, MRI signal characteristics, analysis of associated anomalies in the cord and adjacent structures, as well as in the brain, help determine the most probable diagnostic categories. Identifying some suggestive radiological patterns can lead to a more refined diagnosis, and correlating clinical and biological data can establish the most likely underlying disease. Furthermore, positron emission tomography using [^18^F] fluorodeoxyglucose or [^11^C] methionine may be useful by demonstrating accumulation of these radiotracers in spinal cord tumor cases [104,105].

Current capabilities for tissue characterization in the spinal cord are limited compared to the brain. Advancements in techniques already used in brain MRI may enhance the radiodiagnosis of spinal pathologies. Perfusion-weighted imaging including dynamic-contrast enhanced MRI, detecting increased K^trans^ in spinal cord tumors, may help [106]. Similarly, proton spectroscopy MRI could improve diagnostic specificity by identifying biochemical tissue abnormalities [106]. Diffusion tensor-imaging tractography may offer detailed analyses of white matter tracts, revealing whether a lesion compresses or invades them [5,106]. With the advancement of ultrahigh-field MRI technology, like the 7T MRI, we can hope to achieve unparalleled image clarity, revealing minute details such as changes in specific white matter tracts or the condition of ventral and dorsal nerve roots [107,108]. Techniques such as susceptibility-weighted imaging at 7T may have an important role in the study of spinal cord disease [109].

## 5. Conclusions

Differentiating neoplastic from non-neoplastic spinal cord pathologies is challenging and requires a meticulous radiological analysis. Advancements in MRI techniques may improve the diagnostic capabilities.

## 6. Key Points

Differentiating between neoplastic and non-neoplastic spinal cord pathologies is challenging due to overlapping clinical and radiological features, necessitating a cautious diagnostic approach to avoid unnecessary interventions.

Magnetic Resonance Imaging (MRI), especially T1-weighted and T2-weighted imaging, plays a crucial role in the initial detection of spinal cord lesions, supplemented by other MRI sequences like post-contrast T1WI, T2*-WI, DWI, and FAT-SAT for additional insights.

In situations where MRI is contraindicated, CT myelography can be used as an alternative to identify spinal enlargement and locate lesions. Cerebrospinal Fluid (CSF) analysis also contributes to diagnosing by detecting inflammatory changes or abnormal cells.

Precise localization of the spinal cord lesion is critical, differentiating intra-medullary masses from intradural extramedullary and extradural masses, with the extent of the lesion (longitudinally extensive versus short lesions) offering diagnostic clues.

Assessing lesion signal intensity on MRI, including T1 and T2 signals and features like enhancing areas, cystic components, or diffusion-restricted zones, aids in further characterizing the nature of spinal cord lesions.

Evaluating associated anomalies in spinal MRI, such as spinal central canal enhancement, leptomeningeal enhancement, or bony structures, helps in identifying underlying pathologies.

Assessing potentially associated cerebral abnormalities through MRI can provide additional diagnostic information, helping exclude certain differential diagnoses.

No clinical or radiological signs definitively distinguish between a spinal cord tumor and a pseudotumor.

Spinal cord pseudotumors encompass a wide range of pathologies, with the most common being inflammatory myelitis and degenerative compressive myelopathy.

## Figures and Tables

**Figure 1 cancers-16-01004-f001:**
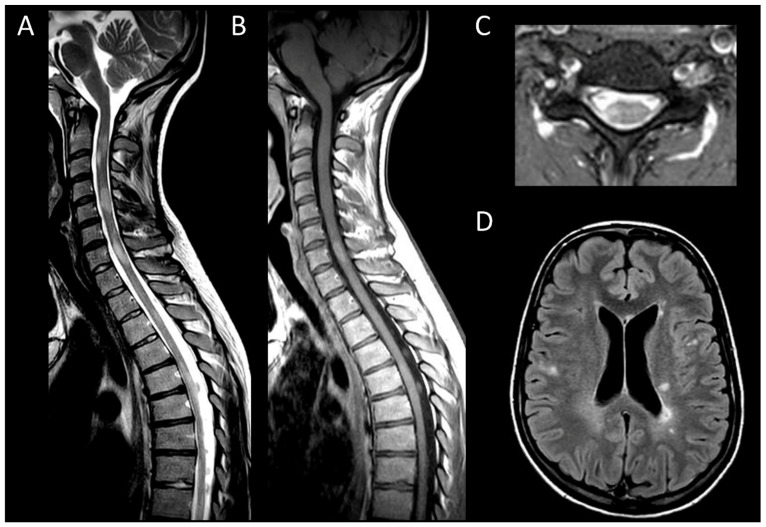
Multiple sclerosis myelitis. (**A**) Sagittal T2-weighted MRI of the cervical spine. (**B**) Sagittal post-contrast T1-weighted MRI of the cervical spine. (**C**) Axial T2-weighted MRI of the cervical spine. (**D**) Fluid-attenuated inversion recovery-weighted MRI of the brain. The figure illustrates multiple short intramedullary myelitis (**A**), including one lesion showing enhancement (**B**) and selectively affecting the right lateral cord (**C**). Brain MRI (**D**) reveals multiple short and ovoid associated lesions in the periventricular and subcortical regions. The imaging data meet the criteria for temporal and spatial dissemination of multiple sclerosis.

**Figure 2 cancers-16-01004-f002:**
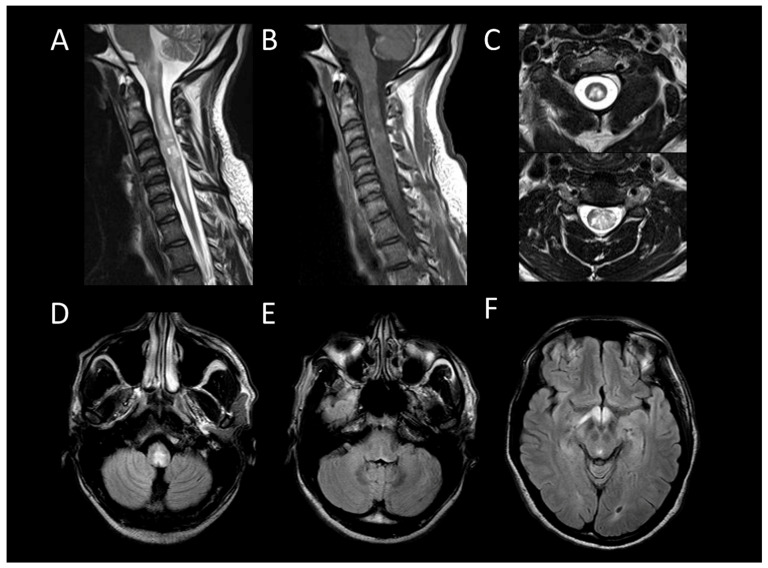
Neuromyelitis optica spectrum disorder positive for aquaporin-4 antibody myelitis. (**A**) Sagittal T2-weighted MRI of the cervical spine. (**B**) Sagittal post-contrast T1-weighted MRI post-injection of the cervical spine. (**C**) Axial T2-weighted MRI of the cervical spine. (**D**–**F**) Fluid-attenuated inversion recovery-weighted MRI of the brain. This figure depicts longitudinally extensive myelitis (**A**) with leptomeningeal enhancement (**B**). Axial images reveal extensive transverse involvement of both white and gray matter with T2 signal lesions as intense as cerebrospinal fluid (bright spotty lesions) (**C**). Brain MRI shows involvement of the area postrema and the optic chiasm (**D**–**F**).

**Figure 3 cancers-16-01004-f003:**
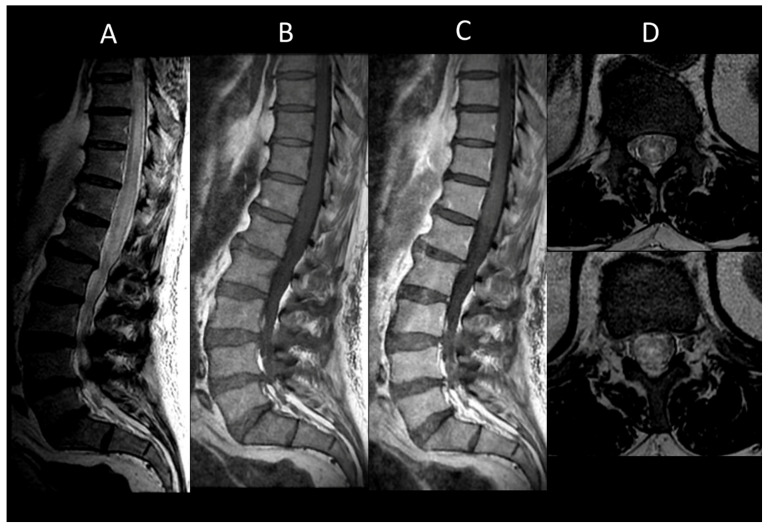
Myelin-oligodendrocyte glycoprotein associated disorder myelitis. (**A**) Sagittal T2-weighted MRI of the thoraco-lumbar spine. (**B**) Sagittal T1-weighted MRI of the thoraco-lumbar spine. (**C**) Sagittal post-contrast T1-weighted MRI of the thoraco-lumbar spine. (**D**) T2-weighted MRI of the thoraco-lumbar spine in sagittal sections. This figure illustrates a longitudinally extensive myelitis of the conus medullaris (**A**), without enhancement (**B**,**C**). On axial images, a transverse myelitis affecting both white and grey matter and a pseudodilatation of the ependymal canal are observed (**D**).

**Figure 4 cancers-16-01004-f004:**
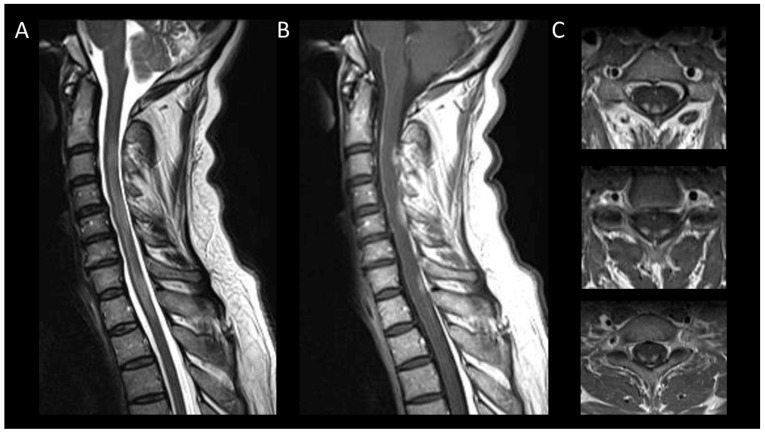
Neurosarcoidosis myelitis. (**A**) Sagittal T2-weighted MRI of the cervical spine. (**B**) Sagittal T1-weighted MRI post-injection of the cervical spine. (**C**) Axial post-contrast T1-weighted MRI of the cervical spine. This figure shows a longitudinally extensive cervical myelitis (**A**) with dorsal enhancement and leptomeningeal enhancement (**B**). The axial images (**C**) demonstrate enhancement of lesions in the posterior subpial white matter.

**Figure 5 cancers-16-01004-f005:**
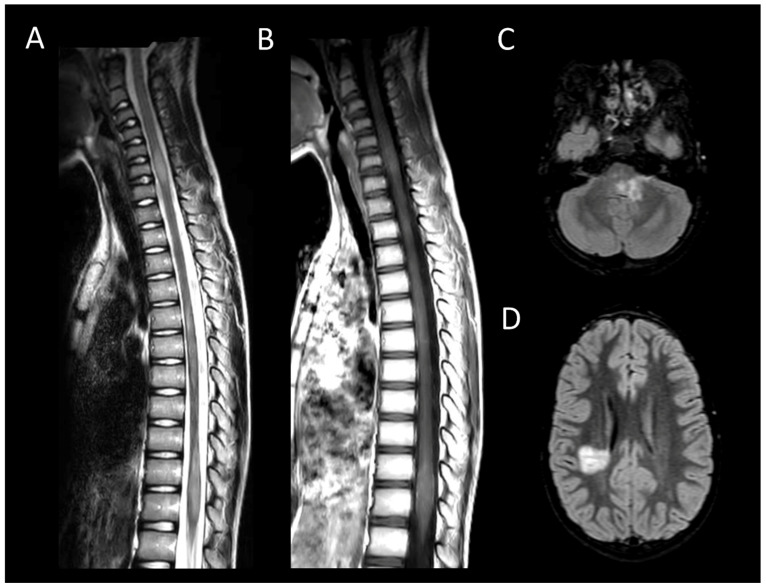
Acute demyelinating encephalomyelitis. (**A**) Sagittal T2-weighted MRI of the spine. (**B**) Sagittal post-contrast T1-weighted MRI of the spine. (**C**,**D**) Fluid-attenuated inversion recovery-weighted MRI of the brain. This figure illustrates multiple short myelitis at the level of the cervical and thoracic spinal cord (**A**), of similar age (not enhanced) (**B**) associated with demyelinating-like lesions of the posterior cranial fossa and periventricular regions (**C**,**D**).

**Figure 6 cancers-16-01004-f006:**
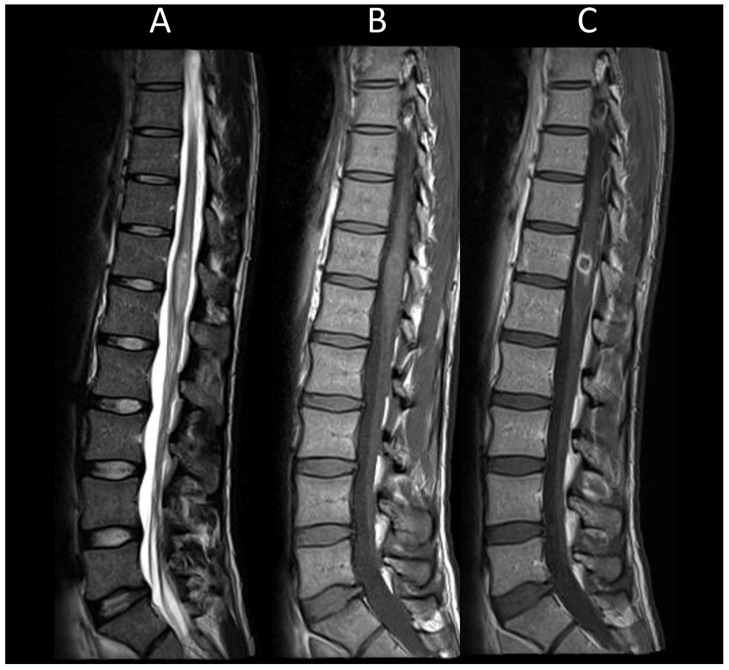
Neuro-Behçet’s myelitis. (**A**) Sagittal T2-weighted MRI of the thoraco-lumbar spine. (**B**) Sagittal T1-weighted MRI of the thoraco-lumbar spine. (**C**) Sagittal post-contrast T1-weighted MRI of the thoraco-lumbar spine. This figure illustrates a myelitis of the conus medullaris (**A**), characterized by ring enhancement and leptomeningeal enhancement (**B**,**C**).

**Figure 7 cancers-16-01004-f007:**
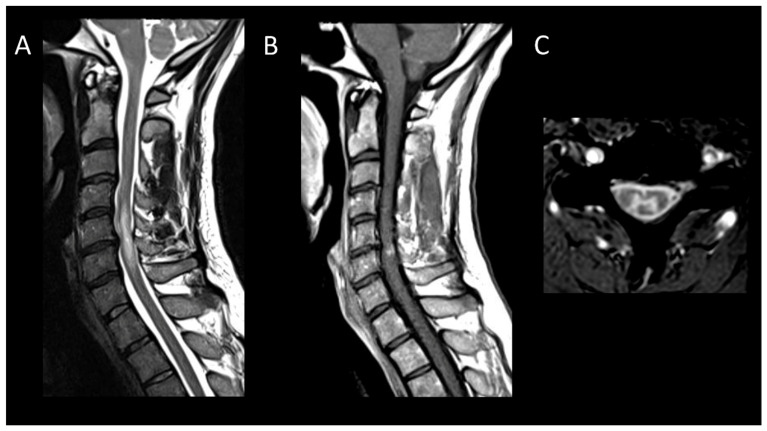
Degenerative compressive myelopathy. (**A**) Sagittal T2-weighted MRI of the cervical spine. (**B**) Sagittal T1-weighted MRI post-injection of the cervical spine. (**C**) Sagittal T2-weighted MRI of the cervical spine in axial sections. This figure illustrates a short myelitis centered on a posterior disc herniation at the level of C5–C6 (**A**) with linear “pancake-like” enhancement at the site of the maximum canal stenosis (**B**). Axial images (**C**) show lesions predominantly affecting gray matter resulting in an H-shaped appearance.

**Figure 8 cancers-16-01004-f008:**
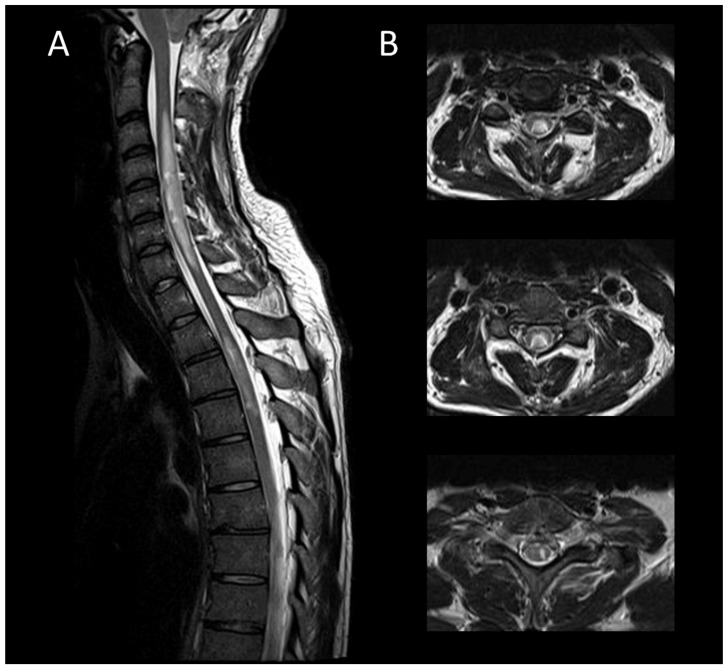
Subacute spinal cord infarction. (**A**) Sagittal T2-weighted MRI of the cervical spine. (**B**) Sagittal T2-weighted MRI of the cervical spine. This figure depicts two spinal cord lesions (one cervical and one thoracic) that are edematous and slightly swollen (**A**), predominantly affecting the gray matter (**B**).

**Figure 9 cancers-16-01004-f009:**
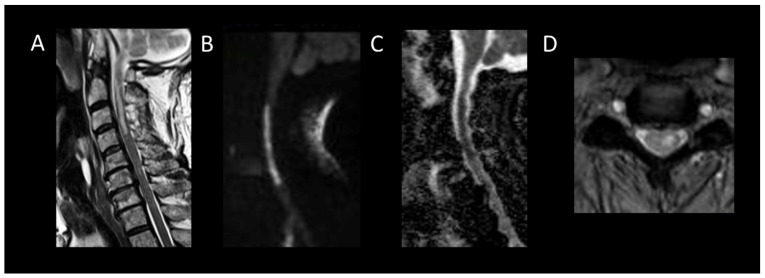
Acute spinal cord infarction. (**A**) Sagittal T2-weighted MRI of the cervical spine. (**B**) Sagittal diffusion-weighted MRI of the cervical spine. (**C**) Sagittal apparent diffusion coefficient map of the cervical spine. (**D**) Axial T2-weighted MRI of the cervical spine. This figure displays a cervical spinal cord lesion (**A**), exhibiting cytotoxic edema (**B**,**C**) and predominantly affecting the gray matter of the left hemicord (**D**).

**Figure 10 cancers-16-01004-f010:**
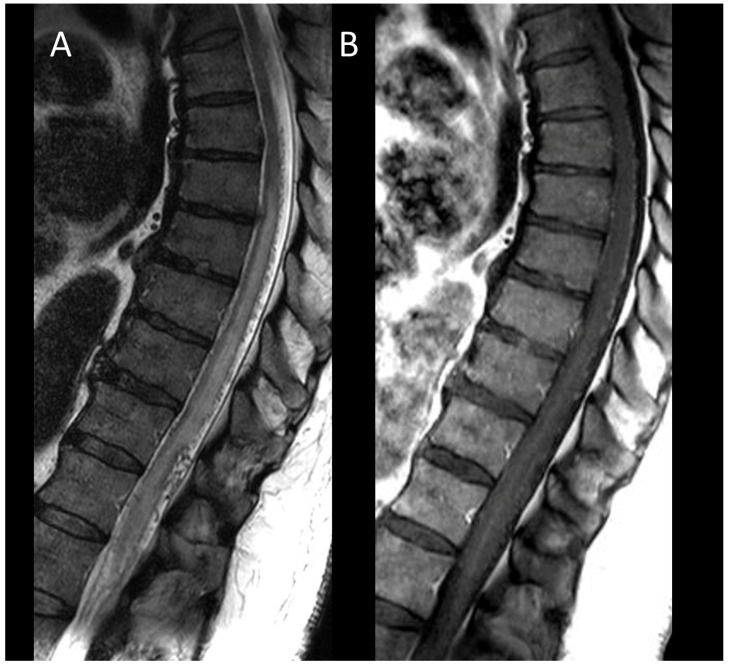
Spinal dural arteriovenous fistula. (**A**) Sagittal T2-weighted MRI of the thoraco-lumbar spine. (**B**) Sagittal post-contrast T1-weighted MRI of the thoraco-lumbar spine. This figure illustrates a longitudinally extensive and swollen edematous lesion of the conus medullaris (**A**) associated with tortuous and dilated perimedullary vessels (**A**).

**Figure 11 cancers-16-01004-f011:**
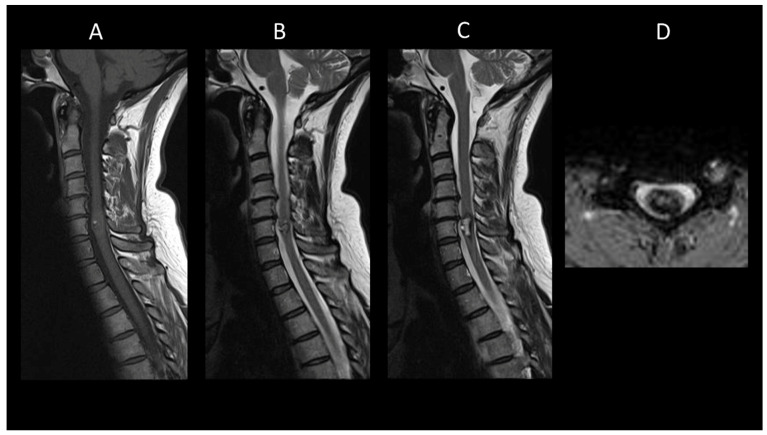
Spinal cavernous malformation. (**A**) Sagittal T1-weighted MRI of the cervical spine. (**B**,**C**) Sagittal T2-weighted MRI of the cervical spinal. (**C**) Axial gradient-recalled echo T2*-weighted imaging of the cervical spine. This figure depicts a moderately swollen, well-defined oval intramedullary lesion with multiple compartments showing variable T1 and T2 signal intensities, resembling a mulberry appearance (**A**,**B**), surrounded by a peripheral hemosiderin rim (**C**). The axial images reveal a blooming artifact on the magnetic susceptibility sequence (**D**).

**Figure 12 cancers-16-01004-f012:**
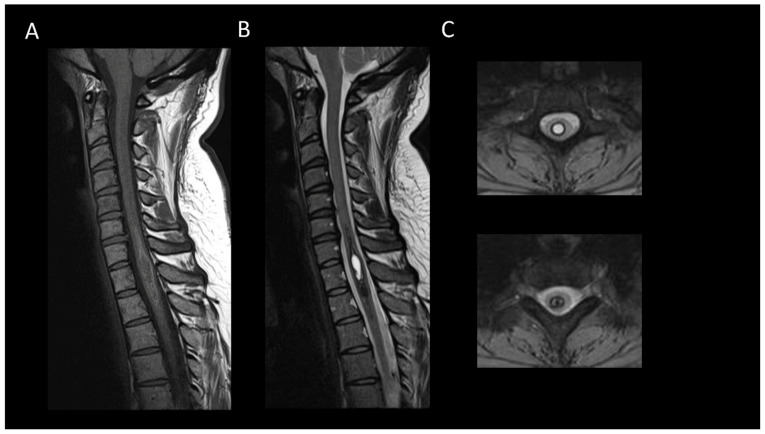
Chronic hematomyelia. (**A**) Sagittal T1-weighted MRI of the cervical spine. (**B**) Sagittal T2-weighted MRI of the cervical spinal. (**C**) Axial T2-weighted MRI of the cervical spinal. This figure displays an intramedullary lesion moderately expanding the spinal cord, appearing isointense on T1-weighted images (**A**) and centrally hyperintense on T2-weighted images (**B**), surrounded by a peripheral ring of hyperintensity on T1 and hypointensity on T2 (**C**).

**Figure 13 cancers-16-01004-f013:**
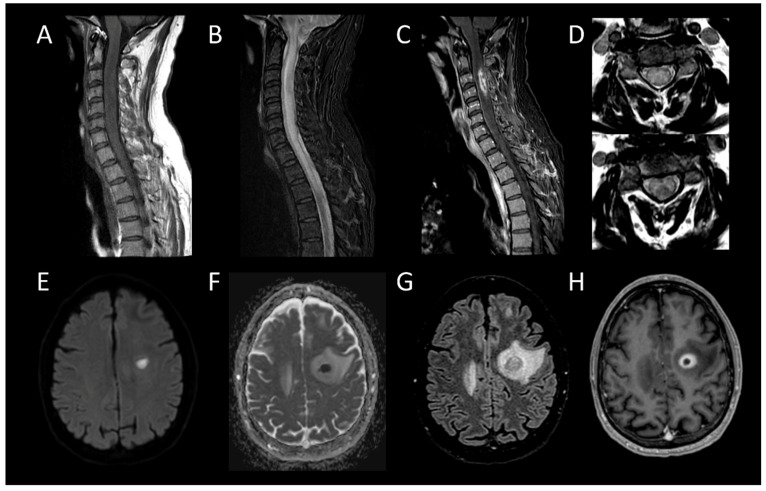
Bacterial Myelitis. (**A**) Sagittal T1-weighted MRI of the cervical spine. (**B**) Sagittal fat-saturated T2-weighted MRI of the cervical spine. (**C**) Sagittal post-contrast T1-weighted MRI of the cervical spinal. (**D**) Axial T2-weighted MRI of the cervical spine. (**E**) Axial diffusion-weighted MRI of the brain. (**F**) Axial apparent diffusion coefficient map of the brain. (**G**) Axial fluid-attenuated inversion recovery-weighted MRI of the brain. (**H**) Axial post-contrast T1-weighted MRI of the brain. This figure illustrates a moderately swollen, longitudinally extensive cervical myelitis (**A**,**B**) with a lesion showing annular enhancement suggestive of an ongoing collection (**C**). Axial images reveal involvement of both white and grey matter (**D**). Cerebral MRI reveals a left frontal lesion with purulent content (**E**,**F**), surrounded by a vasogenic edema zone (**G**), and exhibiting annular enhancement (**H**), indicating an intracerebral abscess.

**Figure 14 cancers-16-01004-f014:**
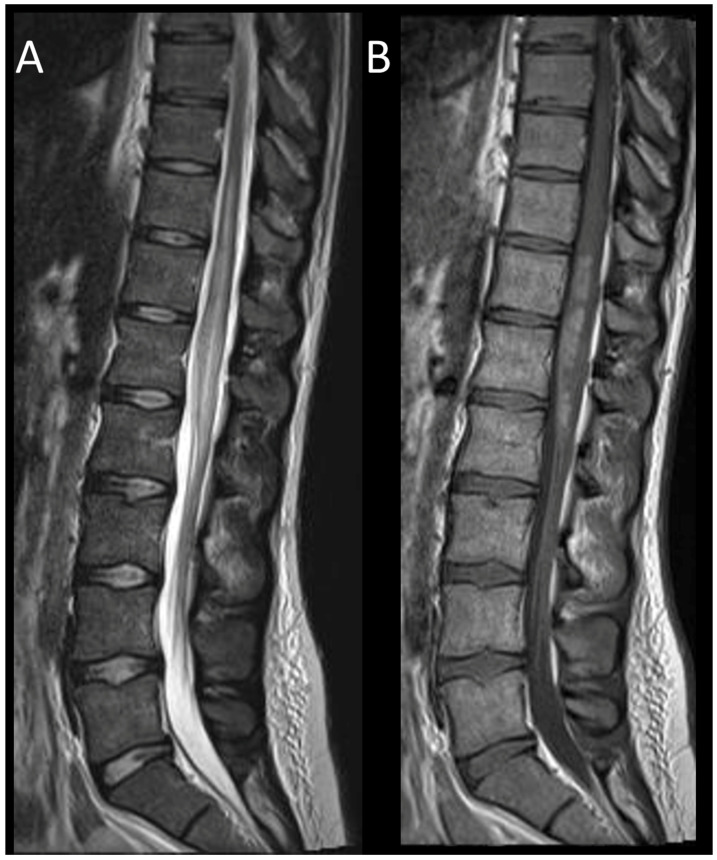
Spinal schistosomiasis. (**A**) Sagittal T2-weighted MRI of the thoraco-lumbar spine. (**B**) Sagittal post-contrast T1-weighted MRI of the thoraco-lumbar spine. This figure depicts a longitudinally extensive myelitis of the conus medullaris (**A**) with patchy and confluent enhancement (**B**).

**Figure 15 cancers-16-01004-f015:**
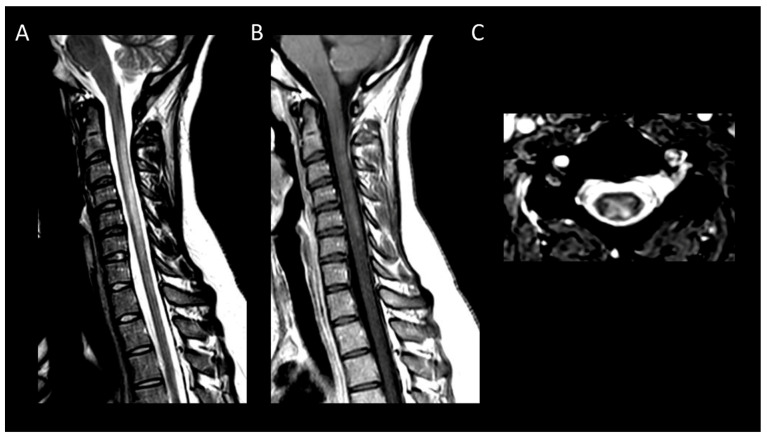
Subacute combined degeneration of the spinal cord. (**A**) Sagittal T2-weighted MRI of the cervical spine. (**B**) Sagittal post-contrast T1-weighted MRI of the cervical spine. (**C**) Axial T2-weighted MRI of the cervical spine. This figure depicts a longitudinally extensive cervical myelitis (**A**) with mild enhancement (**B**) and predominantly involving dorsal columns (**C**).

**Figure 16 cancers-16-01004-f016:**
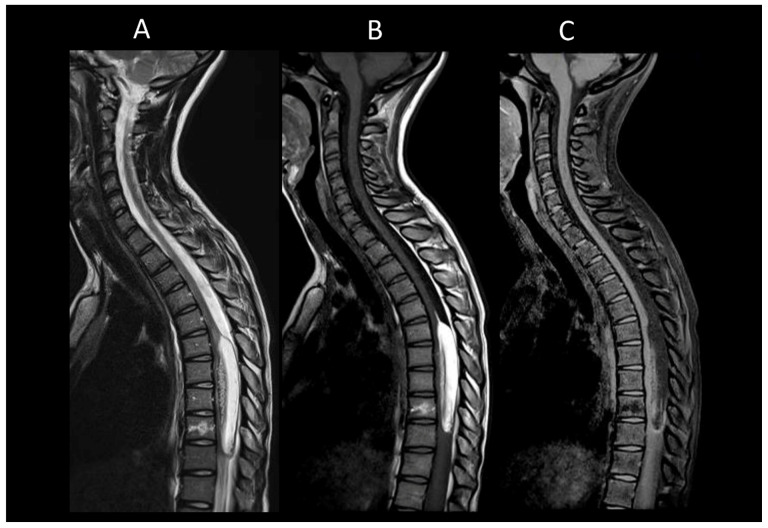
Intra-medullary lipoma. (**A**) Sagittal T1-weighted MRI of the thoraco-lumbar spine. (**B**) Sagittal T2-weighted MRI of the thoraco-lumbar spine. (**C**) Sagittal fat-saturated T1-weighted MRI of the thoraco-lumbar in sagittal section. This figure illustrates an intramedullary lesion, a well-defined lesion exhibiting a fatty signal (**A**–**C**).

**Figure 17 cancers-16-01004-f017:**
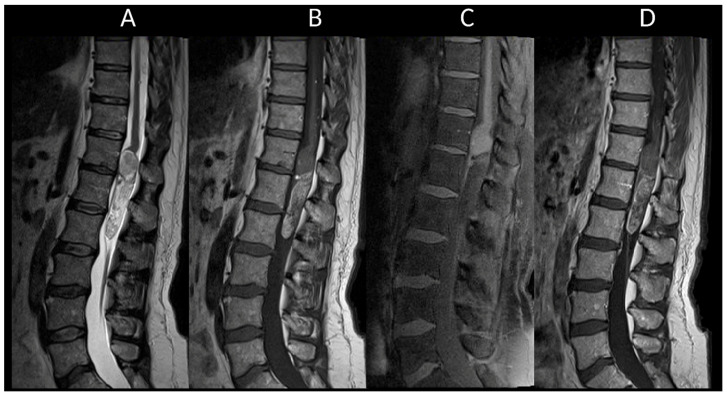
Intra-medullary dermoid cyst. (**A**) Sagittal T1-weighted MRI of the thoraco-lumbar spine. (**B**) Sagittal T2-weighted MRI of the thoraco-lumbar spine. (**C**) Sagittal fat-saturated T1-weighted MRI of the thoraco-lumbar spine. (**D**) Sagittal post-contrast T1-weighted MRI of the thoraco-lumbar spine. This figure shows an intramedullary lesion in the conus region with a cystic appearance containing mixed contents, including a fatty component (**A**–**C**). Note the presence of punctate T1 (**A**) and T2 (**B**) hyperintense signals around the spinal cord, indicating a ruptured dermoid cyst.

**Figure 18 cancers-16-01004-f018:**
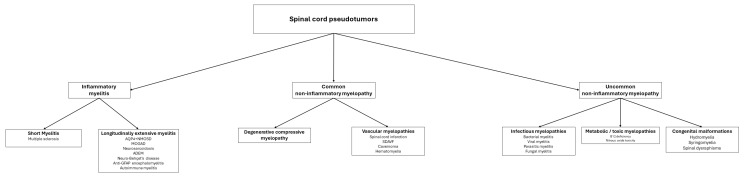
This figure summarizes the main diagnostic ranges to consider when faced with a medullary pseudotumor.

**Table 1 cancers-16-01004-t001:** Main radiological features of spinal cord pseudotumors.

Disease	Long or Short	Typical Location	Enhancement	Characteristic Feature	Associated Spinal Anomalies	Associated Cerebral Anomalies
Multiple Sclerosis	Short	Cervical spinal cord	Non-specific: nodular, patchy, or annular enhancement		Multiple short lesions	Multiple brain lesions in periventricular, subcortical, and posterior fossa regions
Progressive solitary sclerosis	Short	Cervical spinal cord				-
AQP4 + NMOSD	Long	All spinal cord segments	Incomplete ring enhancement	“Bright spotty lesions”		Bilateral optic neuritis,
			Leptomeningeal enhancement			Focal lesions in specific areas: area postrema, midbrain, and diencephalon
						“Pencil-thin” subependymal enhancement
MOGAD	Long	Conus medullaris	Rare	“Ventral sagittal line”		Cerebellar lesions extending to the cerebellar peduncles.
			Leptomeningeal enhancement	“H-shaped”		Uni- or bilateral optic neuritis
				“Pseudodilatation” of the spinal canal		
Neurosarcoidosis	Long	Cervical and high thoracic spinal cord	Dorsal or ventral subpial enhancement	“Trident sign”	Dural masses	Meningeal enhancement of the basal cranium
			Ependymal enhancement			Intra- or extra-axial masses
			Leptomeningeal enhancement			
ADEM	Variable	All spinal cord segments	Inconstant enhancement	Synchronous nature of the lesions		Multiple supratentorial and infratentorial lesions
Neuro-Behçet’s Disease	Variable	All spinal cord segments	Inconstant enhancement	“Bagel-like”		Enhanced lesions in the brainstem, basal ganglia, subcortical matte
						Meningeal enhancement of the basal cranium
Anti-GFAP encephalomyelitis	Long	All spinal cord segments	Central canal and leptomeningeal enhancement	-		T2/FLAIR hyperintensities of the semi-oval centers.
						Radial perivascular enhancement
Auto immune myelitis	Long	All spinal cord segments	Patchy enhancement in about half of the cases	-		
Degenerative compressive myelopathy	Variable	Cervical spinal cord	Inconstant enhancement	“Pancake-like” enhancement	Spinal canal stenosis	
					Degenerative spinal features	
Spinal cord infarction	Variable	All spinal cord segments	Inconstant enhancement	Restricted diffusion	Adjacent vertebral body infarction	
				“Pencil-like”		
				“Owl-eye”		
Spinal dural arteriovenous fistula	Long	Conus medullaris	Inconstant enhancement	Perimedullary dilated vessels		
Spinal cavernous malformation	Variable	Cervico-thoracic spinal cord	No	Mixed signal of blood products		
				Hemosiderin ring		
Hematomyelia	Variable	None	No	None		
Subacute combined degeneration of the spinal cord	Long	Cervical and upper thoracic regions	Inconstant enhancement	“Chevron”		

Abbreviations: ADEM: acute disseminated encephalomyelitis, AQP4: aquaporin-4, GFAP: glial fibrillary acid protein, FLAIR: fluid-attenuated inversion recovery, MOGAD: myelin-oligodendrocyte glycoprotein antibody associated disease, NMOSD: neuromyelitis optica spectrum disorder.

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
