# Peer review of "It Looks Like a Spinal Cord Tumor but It Is Not"

_cancers, 2024, doi:10.3390/cancers16051004_

Round 1
Reviewer 1 Report
Comments and Suggestions for Authors
Fournel et al. have presented a comprehensive review that will serve as an invaluable resource for the radiological differentiation of non-neoplastic spinal cord pathologies, a subject notorious for its complexity and the often necessary collaboration among a multidisciplinary team comprising neuroradiologists, neurologists, and neurosurgeons. It is particularly commendable that the paper was authored by specialists from these diverse fields, enhancing its relevance and applicability in clinical settings. However, several concerns need to be addressed before publication:
Major:
- According to the publisher's description, "Cancers" is a peer-reviewed, open-access journal focused on oncology, published semimonthly online by MDPI. However, this review extensively discusses non-oncological conditions. This discrepancy raises questions about the suitability of this journal for disseminating findings primarily centered on non-cancerous spinal cord pathologies.
- I strongly recommend the inclusion of an additional figure, specifically a diagnostic flowchart, to summarize the content of Chapter 2. This visual representation could greatly enhance comprehension and provide a quick reference for practitioners, facilitating the application of the review's insights into clinical practice.
- Could you specify the clinical data helpful for differentiating between spinal cord tumors and pseudotumors? For instance, should there be a prior administration of steroids, or are results from genetic testing (e.g., for vHL in hemangioblastoma) required? Or maybe no preliminary data is needed? I believe that a section like this may improve communication between neurologists and radiologists.
- The authors should try to suggest timing for control MRI scans. In what situations is it appropriate to wait for 6 to 12 months before conducting a follow-up MRI, and when should one exercise greater caution (e.g., in cases of ISCA?)? Offering clear recommendations on this matter would aid in decision-making and patient management, particularly in navigating the balance between immediate intervention and watchful waiting.
Minor:
- The inconsistency in the use of Roman and Arabic numerals (e.g., II. 1, II.2, ...) throughout the document appears unprofessional. A standardized format for section headings and subpoints would improve the clarity and aesthetic appeal of the manuscript.
- Attention should be given to eliminating formatting errors, such as double spaces (e.g. line 35) and unnecessary hyphens (e.g. line 49).
- The paper should better discuss the utility of additional MRI sequences (e.g., SWI in vascular lesions?)
- The statement "Rare in adults, intraspinal abscesses are typically linked to bacterial meningitis, spondylodiscitis, or infective endocarditis [90]" might imply that intraspinal cord abscesses (ISCA) are more common in children. However, this contradicts the findings from a systematic review in 2022, which identified only 64 pediatric ISCA cases across 58 papers, compared to a study that reported 70 adult ISCA cases from January 1949 to May 2022. This discrepancy should be clarified to avoid misinterpretation of ISCA prevalence across different age groups.
- In which patients fungal myelitides can be expected? Which clinical information may be important for radiologists in patients with fungal myelitides suspiction?
Author Response
Dear Reviewer,
Thank for your comment.
Major:
- According to the publisher's description, "Cancers" is a peer-reviewed, open-access journal focused on oncology, published semimonthly online by MDPI. However, this review extensively discusses non-oncological conditions. This discrepancy raises questions about the suitability of this journal for disseminating findings primarily centered on non-cancerous spinal cord pathologies.
- Indeed, this article focuses on non-tumoral pathologies. However, we believe that understanding non-tumoral pathologies that mimic spinal cord tumors is important for specialists managing spinal cord tumors, as it may help to avoid invasive procedures.
- I strongly recommend the inclusion of an additional figure, specifically a diagnostic flowchart, to summarize the content of Chapter 2. This visual representation could greatly enhance comprehension and provide a quick reference for practitioners, facilitating the application of the review's insights into clinical practice.
- We have added the Figure 19 which summarizes main causes of spinal cord pseudotumors in the revised manuscript.
- Could you specify the clinical data helpful for differentiating between spinal cord tumors and pseudotumors? For instance, should there be a prior administration of steroids, or are results from genetic testing (e.g., for vHL in hemangioblastoma) required? Or maybe no preliminary data is needed? I believe that a section like this may improve communication between neurologists and radiologists.
- Indeed, clinical data can contribute to diagnosis. According to the literature, patients with neurological pathologies have shorter symptom durations compared to patients with spinal cord tumors. In certain situations, a trial treatment with corticosteroids may be beneficial. We have specified these points in the revised manuscript: "From a clinical perspective, this study indicated that patients with neurological diseases had significantly shorter symptom duration than those with neoplasms [43]. In patients presenting with acute spinal cord swelling, corticosteroid therapy could help differentiate pseudo-tumors from true tumors, although infection must be ruled out beforehand [42]"
- The authors should try to suggest timing for control MRI scans. In what situations is it appropriate to wait for 6 to 12 months before conducting a follow-up MRI, and when should one exercise greater caution (e.g., in cases of ISCA?)? Offering clear recommendations on this matter would aid in decision-making and patient management, particularly in navigating the balance between immediate intervention and watchful waiting.
- When there is an atypical clinical presentation or an acute onset, we usually perform an MRI follow-up at 6 weeks because spinal cord lesions tend to change rapidly in cases of neurological pathologies. We have specified this point in the revised version of the manuscript: "Additionally, early MRI monitoring at 6 weeks could be beneficial in these patients or in cases of atypical clinical presentation."
Minor:
- The inconsistency in the use of Roman and Arabic numerals (e.g., II. 1, II.2, ...) throughout the document appears unprofessional. A standardized format for section headings and subpoints would improve the clarity and aesthetic appeal of the manuscript.
- In the revised manuscript, we have standardized the use of Roman numerals for all sections
-
Attention should be given to eliminating formatting errors, such as double spaces (e.g. line 35) and unnecessary hyphens (e.g. line 49).
- We have corrected the formatting errors in the revised manuscript.
-
The paper should better discuss the utility of additional MRI sequences (e.g., SWI in vascular lesions?)
- Currently, advanced MRI techniques are not widely used except for diffusion in exploring spinal pathologies. However, SWI is indeed a promising technique for ultra-high field MRI. We have included this point in the revised manuscript: "Techniques such as susceptibility-weighted imaging at 7T may have an important role in the study of spinal cord disease [111]."
- The statement "Rare in adults, intraspinal abscesses are typically linked to bacterial meningitis, spondylodiscitis, or infective endocarditis [90]" might imply that intraspinal cord abscesses (ISCA) are more common in children. However, this contradicts the findings from a systematic review in 2022, which identified only 64 pediatric ISCA cases across 58 papers, compared to a study that reported 70 adult ISCA cases from January 1949 to May 2022. This discrepancy should be clarified to avoid misinterpretation of ISCA prevalence across different age groups.
- We have revised this sentence accordingly: "Rare in both adults and children"
- In which patients fungal myelitides can be expected? Which clinical information may be important for radiologists in patients with fungal myelitides suspiction?
- Fungal mylitis primarily affects immunocompromised patients. We have revised the manuscript accordingly: "Fungal myelitis is a rare condition that primarily affect immunocompromised patients."
Reviewer 2 Report
Comments and Suggestions for Authors
This review article presents and indicates MRI findings in various diseases of the spinal cord, mainly in view of differentiating between neoplastic and non-neoplastic diseases. Precise and detailed explanation could be educational for many readers. Each figure is almost enough to show their MRI findings.
But, its title “It looks like … Is Not” does not include “MRI”. There are not findings of other modalities at all. Especially, PET using FDG, Methionine, or other amino acid tracers could be useful to differentiate between them. In discussion, those descriptions can be added. Moreover, for example, permeability imaging using dynamic MRI can differentiate between them.
Author Response
Dear Reviewer,
Thank you for your comments
We revised the manuscript to mention the utility of positron-emission tomography in the discussion section
"Furthermore, positron emission tomography using [18F] fluorodeoxyglucose or [11C] methionine may be useful by demonstrating accumulation of these radiotracers in spinal cord tumor cases [106,107]"
We have also clarified the utility of perfusion-weighted imaging:
"Perfusion-weighted imaging including dynamic-contrast enhanced MRI, detecting increased Ktrans in spinal cord tumors, may help [108]. "
Round 2
Reviewer 1 Report
Comments and Suggestions for Authors
The authors have successfully addressed all of my concerns. The paper can be published in its current version. Congratulations on your outstanding work!